# Antiarrhythmic and Inotropic Effects of Selective Na^+^/Ca^2+^ Exchanger Inhibition: What Can We Learn from the Pharmacological Studies?

**DOI:** 10.3390/ijms232314651

**Published:** 2022-11-24

**Authors:** Norbert Nagy, Noémi Tóth, Péter P. Nánási

**Affiliations:** 1ELKH-SZTE Research Group of Cardiovascular Pharmacology, 6720 Szeged, Hungary; 2Department of Pharmacology and Pharmacotherapy, Albert Szent-Györgyi Medical School, University of Szeged, 6720 Szeged, Hungary; 3Department of Physiology, Faculty of Medicine, University of Debrecen, 4032 Debrecen, Hungary; 4Department of Dental Physiology and Pharmacology, Faculty of Dentistry, University of Debrecen, 4032 Debrecen, Hungary

**Keywords:** heart, Na^+^/Ca^2+^ exchanger, arrhythmia, Ca^2+^-handling, inotropy

## Abstract

Life-long stable heart function requires a critical balance of intracellular Ca^2+^. Several ion channels and pumps cooperate in a complex machinery that controls the influx, release, and efflux of Ca^2+^. Probably one of the most interesting and most complex players of this crosstalk is the Na^+^/Ca^2+^ exchanger, which represents the main Ca^2+^ efflux mechanism; however, under some circumstances, it can also bring Ca^2+^ into the cell. Therefore, the inhibition of the Na^+^/Ca^2+^ exchanger has emerged as one of the most promising possible pharmacological targets to increase Ca^2+^ levels, to decrease arrhythmogenic depolarizations, and to reduce excessive Ca^2+^ influx. In line with this, as a response to increasing demand, several more or less selective Na^+^/Ca^2+^ exchanger inhibitor compounds have been developed. In the past 20 years, several results have been published regarding the effect of Na^+^/Ca^2+^ exchanger inhibition under various circumstances, e.g., species, inhibitor compounds, and experimental conditions; however, the results are often controversial. Does selective Na^+^/Ca^2+^ exchanger inhibition have any future in clinical pharmacological practice? In this review, the experimental results of Na^+^/Ca^2+^ exchanger inhibition are summarized focusing on the data obtained by novel highly selective inhibitors.

## 1. Introduction

Calcium ions (Ca^2+^) play a pivotal role in several aspects of cardiac function. They contribute to the long-lasting plateau phase of the ventricular action potential, trigger the opening of the ryanodine receptors (RyRs), contribute to the feed-back regulation of Ca^2+^-influx, induce the activation of the contractile machinery of myocytes, and, finally, are involved in the regulation of many cardiac signal transduction pathways. Under physiological conditions, cardiac cells are in equilibrium, i.e., the entry and efflux of Ca^2+^ are equal on a long time scale. This is also the case in the diseased heart; however, equilibrium can be reached only at a higher cytosolic Ca^2+^ level. This leads to cardiac Ca^2+^ overload, which may be considered as adaptation to the compromised contractility, but also results in the impaired relaxation and development of life-threatening cardiac arrhythmias [1,2,3].

There are two major strategies applied to treat insufficient cardiac contractility. The first approach is to increase the Ca^2+^ content of the myocytes. This classical strategy is primarily based on the elevation of the cytosolic Na^+^ concentration—either by a partial suppression of the Na^+^/K^+^ pump (cardiac glycosides, e.g., digoxin) or by increasing the amplitude of the late Na^+^ current. In both cases, the increased intracellular Na^+^ activity results in elevation of the intracellular Ca^2+^ concentration via the Na^+^/Ca^2+^ exchanger (NCX). Unfortunately, both strategies may also carry serious proarrhythmic risk. This is especially true for cardiac glycosides, which can be applied in clinical practice only cautiously due to their very narrow therapeutic window. The use of digoxin is well-explained in heart failure (HF) patients with reduced ejection fraction; however, the supporting evidence is conflicting [4]. A recent systematic review and meta-analysis reported that digoxin has no effect on the all-cause mortality and hospitalization in heart failure with preserved ejection fraction patients [5]. Furthermore, a recent study showed poor outcomes in atrial fibrillation patients when digoxin was used instead of beta-blockers [6]. Regarding the late Na^+^ current, it is a prospective target of suppression rather than stimulation due to its strong proarrhythmic activity [7,8].

An alternative possibility to augment the developed force without increasing the Ca^2+^ load in cardiac cells is the application of calcium sensitizers. The most commonly used of these drugs is levosimendan, which also increases the cAMP level in cardiac cells because of its moderate phosphodiesterase inhibitory effect [9,10,11]. However, this seems to be a useful combination since both actions are known to increase contractility. A recently-developed agent, omecamtiv mecarbil, acts exclusively on the contractile machinery by increasing its Ca^2+^-sensitivity [12,13]. It is important to bear in mind that shifting the contractile threshold to lower cytosolic Ca^2+^ concentrations may cause difficulties in relaxation (i.e., diastolic stiffness).

## 2. Intracellular Ca^2+^ Handling in the Heart

In cardiac myocytes, the intracellular Ca^2+^ level is tightly controlled by several mechanisms, including excitation–contraction coupling (ECC). The main function of the coupling is to mediate between the membrane potential change and cardiac contraction. The E–C coupling is initiated by the opening of L-type Ca^2+^ channels, activated in the initial phase of the action potential, and its open state is long enough—mainly during the plateau phase of the action potential—to provide sufficient Ca^2+^ influx to the cell [1]. L-type Ca^2+^ channels are expressed abundantly in the membrane invaginations, called T-tubules, being closely associated with the RyRs located in the membrane of the sarcoplasmic reticulum (SR). It is important to note that there is no direct protein–protein interaction between L-type Ca-channels and RyRs; a narrow gap exists between these structures, called as “fuzzy-space”, where the intracellular Ca^2+^ concentration may reach a considerably higher level than in the bulk space of the cytosol [14]. Ca^2+^, entering through L-type Ca^2+^ channels, open RyRs, leading to the release of large amounts of the stored Ca^2+^ from the SR and ultimately resulting in the development of the intracellular Ca^2+^-transient. The released Ca^2+^ binds to the myofilaments, specifically to troponin-C, leading to cardiac contraction. After contraction, the relaxation process requires sequestration of the released Ca^2+^ back into the SR. This is achieved by the SR Ca-ATPase (SERCA), while a smaller fraction is extruded from the cell by the Na^+^/Ca^2+^ exchanger (NCX), located in the sarcolemma. The activity of the SERCA is controlled by phospholamban (PLB), which inhibits the SERCA when tightly associated. Phosphorylation of PLB, e.g., as a result of beta-adrenergic activation, leads to dissociation of PLB with the concomitant improvement of SERCA activity [1] (Figure 1).

Under normal conditions, the Ca^2+^ released from the intracellular store is initiated by Ca^2+^ ions that trigger the ryanodine receptors. The source of the trigger Ca^2+^ could be mainly the L-type Ca^2+^ channels; however, further Ca^2+^-sources (such as reverse mode of the NCX) could also contribute [1]. It must be noted, however, that trigger Ca^2+^ contributes also directly to the activation of myofilaments in a species-dependent manner. The ratio of Ca^2+^ influx and the amount of released Ca^2+^ gives the amplification of Ca^2+^ release, called as the gain of E–C coupling.

The ryanodine receptors are organized in clusters and an elementary Ca^2+^ release from these clusters is the so-called Ca^2+^ spark. The *spontaneous Ca^2+^ sparks* do not depend on extracellular Ca^2+^ entry through the L-type Ca^2+^ channels [15,16]: spontaneous sparks were found in the absence of extracellular Ca^2+^, during pharmacological block of the I_CaL_, and even in saponin-permeabilized cells [15,16,17,18]. It is feasible that spontaneous Ca^2+^ sparks are influenced by intracellular Ca^2+^ and intra-luminal Ca^2+^ level. The *evoked sparks* are ignited by Ca^2+^-induced Ca^2+^ release via L-type Ca^2+^ channels, and it seems that their kinetics, amplitude, and spatial properties are similar to the spontaneous sparks.

It was demonstrated that the evoked sparks can be attributable to the Ca^2+^ flux [19]. The discovery of Ca^2+^ sparks intimately changed the understanding of the nature of Ca^2+^ transients: the Ca^2+^ release from the SR consists of “discrete units,” rather than being a continuum [20] that provides graded control of Ca^2+^ release, i.e., E–C coupling can be changed by altering the number of recruited release units [15,21].

## 3. Physiology and Pharmacology of the Exchanger

NCX belongs to the Ca^2+^/cation antiporter superfamily, having the isoforms NCX1, NCX2, and NCX3 encoded by genes of SLC8A1-SLC8A3, where the NCX1 splice variants represent the cardiac isoform of NCX (for detailed molecular biological description see reference [22]).

Normally, NCX extrudes 3 Na^+^ for 1 Ca^2+^ (*forward mode operation*), generating electrogenic transport. In this case, inward current is generated while outward current is generated during the *reverse mode function*, when Ca^2+^ entry is combined with Na^+^ efflux (Figure 2).

The direction of the NCX current is regulated by intracellular and extracellular Na^+^ and Ca^2+^ levels as well as the actual transmembrane potential. Among these, the level of the intracellular Ca^2+^ and membrane voltage are altered within a wide range from beat to beat, causing fluctuation of the reversal potential of NCX in each cycle. Reverse mode activity of NCX develops when the reversal potential of NCX is more positive than the membrane potential, while forward mode operation is anticipated if the reversal potential is more negative than the membrane potential [23].

Since the estimated reversal potential of NCX ranges from −30 to −40 mV in the presence of a diastolic Ca^2+^ level, both reverse and forward mode functions are feasible during each action potential [1]. However, there is no agreement regarding the timing of the reversal point or the ratio between the two operational modes, neither in modelling studies nor in experimental results. The direction of the transport has crucial importance regarding the effect of NCX inhibition.

### 3.1. NCX Current during an Action Potential

Egan and Twist (1989) described NCX current as an inward current during the action potential, calculated from experimental results [24]. In line with this, duBell et al. [25] observed inward NCX current which prolonged action potential duration in rats. Noble et al., in 1991, predicted that NCX current is inward during the action potential, with a very brief reverse mode following the action potential upstroke [26]. The Luo–Rudy model (1994) predicted that NCX current operates in reverse mode in the first 100 ms of the guinea pig action potential then turns to forward mode, reaching its peak during terminal repolarization [27]. In this study, however, the authors calculated only with a peak free [Ca^2+^]_i_ of 1 µmol/L. Experimental data from Grantham and Cannell (1996) obtained in guinea pig myocytes showed that NCX current is mainly reversed during the action potential and reaches the reversal point shortly after complete repolarization [28]. Using an improved model, Noble et al., in 1998, observed that the time of NCX reversal is approximately 200 ms after the upstroke in guinea pigs [29]. Faber and Rudy, in 2000, described a small outward current during the plateau phase of the action potential which turned to forward immediately after terminal repolarization. Furthermore, they pointed out that NCX was very sensitive to increases in [Na^+^]_i_, causing considerable changes in both the reverse and forward modes, and the former may contribute to frequency-dependent shortening of the action potential [30]. The Bers group found that NCX current is mainly forward during the action potential in rabbits, with an estimated reversal point of less than 30 ms following the upstroke because of the very high Ca^2+^ level in the subsarcolemmal space [31]. This sensitivity of NCX current to membrane potential changes was, later, further supported by Armoundas et al. in 2003 [32].

The first experimental evidence applying a relatively selective NCX inhibitor SEA-0400 (3 µM) was published by Bányász et al. in 2012 in guinea pig ventricular myocytes [33]. They found NCX current to be completely inward during the action potential, reaching its peak at the end of the plateau phase. Later, Nagy et al. (2014) reported similar results by applying ORM-10103, a more selective inhibitor without an effect on I_CaL_. The authors reported a small and brief outward current corresponding to phase 1 repolarization and a small inward current in the entire action potential. Both the reverse and forward mode operation of NCX increased when intracellular Na^+^ concentration was increased by activation of late sodium current [34]. Horvath et al. demonstrated an entirely inward NCX current, defined as an ORM-10962 sensitive current using the so-called “onion-peeling technique” in canine ventricular myocytes [35].

However, the time course of NCX current during the action potential is not completely consistent, it seems feasible that experimental data and modelling predictions suggestonly marginal reverse NCX current during the action potential under normal conditions. This means that selective inhibition may influence mainly forward mode operation of NCX, resulting in reduced Ca^2+^ extrusion from the intracellular space with concomitant suppression of the NCX-induced inward current during the action potential, which might tend to shorten the action potential (Figure 3).

### 3.2. Pharmacology of Novel NCX Inhibitors

As previous NCX inhibitors (such as KB-R7943) exerted non-selective behaviour, the interpretation of the results was difficult [36]. Especially, the results were contaminated by the parallel I_CaL_ block, since it can markedly influence the Ca^2+^ homeostasis, distorting the effect of selective NCX inhibition. Furthermore, it may decrease the intracellular Ca^2+^ content that counteracts the possible positive inotropic effect of the selective NCX inhibition. Estimation of the time course of NCX current is also difficult, as I_CaL_ overlaps the reverse NCX current during an action potential. Therefore, it was a large demand from the researchers to drug companies to develop new, completely selective NCX inhibitor compounds.

SEA-0400 (2-(4-(2,5-difluorobenzyloxy)phenoxy)-5-ethoxyaniline) was the first NCX inhibitor which was shown to exert improved selectivity. The EC_50_ of the NCX block was 3.35 ± 0.82 µM, for forward mode, and 4.74 ± 0.69 µM for reverse mode operation [36]. SEA-0400 did not inhibit K^+^ or Na^+^ currents; however, it decreased I_CaL_ with EC_50_ of 3.6 µM. This means that the typically used 1 µM concentration causes approximately a 20% reduction in I_CaL_ amplitude. Still, the submicromolar concentrations of SEA-0400 could be considered relatively selective for NCX [36].

ORM-10103, a novel compound with improved selectivity, was investigated in detail in 2013 [37]. The major benefit of ORM-10103 is the lack of effect on I_CaL_; however, a 20% blocking effect on I_Kr_ was identified. ORM-10103 equally inhibited both the reverse (EC_50_: 960 nM) and the forward (EC_50_: 780 nM) modes of NCX. Importantly, I_K1_, I_Ks_, I_to_, and Na/K pump were not influenced by 3 µM ORM-100103.

The next compound, ORM-10962, was further improved, displaying complete selectivity to NCX. ORM-10962 equally inhibited both the reverse and forward mode NCX currents with considerably lower EC_50_ values (outward: 67 nM, inward: 55 nM) [38].

### 3.3. Potential Therapeutic Benefit of Pharmacological NCX Inhibition

During ECC, a stable Ca^2+^ balance is crucial for the Ca^2+^ handling integrity; therefore, the Ca^2+^ influx and Ca^2+^ efflux must be equal, causing no Ca^2+^ gain or loss within beats [2,39].

In several diseases, the intracellular Ca^2+^ largely increases and considerably facilitates the activity of the forward NCX [40,41]. The Ca^2+^ removal by the NCX generates inward Na^+^ current that could lead to membrane depolarization causing triggered activity either during phase 2 and phase 3 of repolarization (early afterdepolarization, EAD) or at resting membrane potential (delayed afterdepolarization, DEAD). These triggered activities can lead to the development of extrasystole and—in the presence of arrhythmogenic substrate—may cause life threatening arrhythmias. Additionally, the actual intracellular Na^+^ level also significantly alters the function of NCX. Intracellular Na^+^ concentration can be increased due to suppressed Na^+^/K^+^ ATPase activity, increased Na^+^/H^+^ function, or via enhanced late Na^+^ current during the action potential plateau. The increased Na^+^ level activates the reverse mode of NCX, causing net Ca^2+^ entry, which, in turn, facilitates the activity of the forward mode leading to development of triggered activity [40,41]. Therefore, theoretically, both the inhibition of the forward mode (due to suppressed depolarizing inward current) and inhibition of the reverse mode (due to decreased Ca^2+^ entry that causes secondary reduction in the forward mode) could be antiarrhythmic [42].

Another aspect of the selective NCX inhibition is the proposed positive inotropic effect. Considering that the contribution of the reverse mode during the action potential is restricted, NCX inhibition mainly prevailed on the forward mode. The decreased Ca^2+^-extrusion could increase the intracellular Ca^2+^ level, which is able to accelerate the Ca^2+^-dependent inactivation of the L-type Ca^2+^ channels to match the Ca^2+^-influx to the Ca^2+^ efflux. Therefore, presumably, NCX inhibition is able to set a new Ca^2+^ balance where NCX activity is reduced and intracellular Ca^2+^ is elevated, leading to a consequent increase in contractility.

## 4. Effect of Novel NCX Inhibitors on Cellular Arrhythmogenic Mechanisms

In most cases, cardiac arrhythmias are complex phenomena involving several cellular mechanisms at the same time, such as EAD, increased repolarization dispersion, Ca^2+^ overload and DAD, alternans, depolarized membrane potential, shift in the refractory period, etc. Furthermore, arrhythmia experiments could also differ in experimental conditions. Accordingly, Ca^2+^ overload-related arrhythmias could be caused by Na^+^-induced Ca^2+^ overload following Na/K ATPase blockade, or may be due to increased sympathetic tone, hypokalaemia, etc. In this part of the review, the effects of selective NCX inhibition on elementary arrhythmogenic events are summarized. Since, among NCX inhibitors, only the SEA-0400 and ORM-compounds can be considered more or less selective, the next sections are restricted to summarizing the results obtained with these inhibitors.

### 4.1. DAD and Ca^2+^ Waves

The first observation of DAD-related events was reported by Segers in 1940 and 1947. In frog, turtle, and rabbit hearts, spontaneous activity was observed which could be facilitated by external Ba^2+^ or increased external Ca^2+^. The phenomenon was termed as “potentiels tardifs,” or late potentials. The first direct measurements of DADs were carried out in 1977 by Crenfield et al. [43]. It was found that intracellular Ca^2+^ content can be increased via increased sympathetic tone, or can develop as a consequence of several diseases, such as catecholaminergic polymorph tachycardia, hyperthyreosis, etc., when the cells are overloaded with Ca^2+^. Ca^2+^ overload increases Ca^2+^ reuptake to the SR leading to spontaneous diastolic Ca^2+^ releases. These events drive the forward mode of the Na^+^/Ca^2+^ exchanger, causing spontaneous depolarizations [44].

Since NCX function is considered to be a crucial player of the DAD development, selective NCX inhibition was a promising candidate to suppress arrhythmias related to DADs [42]. SEA-0400 was reported to effectively suppress arrhythmogenic afterdepolarizations under pathophysiological conditions associated with Na^+^- and the consequent Ca^2+^ overload [45,46,47]. In these studies, intracellular Na^+^ gain was induced by oubain/strophantine administration in guinea pigs and dogs.

Fujiwara et al. showed a close link between spontaneous Ca^2+^ releases, DADs, and extra beats following rapid pacing [48]. In this study, SEA-0400 suppressed the triggered activity by attenuating DADs but could not prevent the spontaneous Ca^2+^ waves.

Among new NCX inhibitors, the effect of ORM-10103 was tested against DAD development [37]. DAD was evoked by 40 consecutive stimuli, having a cycle length of 400 ms on canine right ventricular Purkinje fibres in the presence of 0.2 µM strophantin. Furthermore, 3 and 10 µM ORM-10103 decreased the amplitude of DADs in a concentration-dependent manner.

In a further study, strophantin-induced spontaneous Ca^2+^ releases were investigated in isolated dog ventricular myocytes during rest after a rapid pacing period. In the pacing-free period, several spontaneous Ca^2+^ releases were observed which were significantly suppressed by 10 µM ORM-10103. In this paper, the effect of strophantin on steady-state Ca^2+^ transients was also investigated: 1 µM strophantin markedly increased the amplitude of Ca^2+^ transients; however, when the cells were pre-treated with 10 µM ORM-10103, strophantin failed to increase their amplitude [34].

Similar results were obtained by using ORM-10962 [38]. Dog right ventricular Purkinje fibres were treated with 150 nM digoxin and were stimulated at a 400 ms pacing cycle length. It was observed that 1 µM ORM-10962 significantly suppressed the amplitude of DADs. When the reverse mode NCX was facilitated by using 70 mM external NaCl, the subsequently applied 1 µM ORM-10962 reduced the amplitude of the Ca^2+^ transient [38,49]. The application of reduced external Na^+^ shifted the NCX reversal potential toward more negative values and increased the time for reverse mode operation.

These results suggest that selective NCX inhibition effectively suppresses the occurrence of the DADs. Regarding the mechanism, it is feasible that DAD can be suppressed either directly, i.e., by inhibition of the forward mode, or indirectly via reduction of reverse mode leading to decreased intracellular Ca^2+^ concentration and reduced forward mode NCX activity. Since, in these experiments, the DADs were evoked by Na/KATPase inhibition, it is possible that the increased intracellular Na^+^ facilitated the reverse mode activity of NCX, resulting in increasedCa^2+^ entry. This scenario is further supported by the fact that NCX inhibition prevented Ca^2+^ accumulation caused by strophantin, while the magnitude of the Ca^2+^ transient was unchanged [34]. Therefore, it seems feasible that ORM-compounds suppress DADs and spontaneous Ca^2+^ releases via reverse mode NCX inhibition during Na^+^-induced Ca^2+^ overload.

Ca^2+^ waves induced by rapid pacing in dog atrium were found to activate the forward mode of NCX and markedly depolarized the membrane. The application of SEA-0400 or ORM-10103 effectively abolished the Ca^2+^ waves [50].

It appears that selective NCX inhibition displays a beneficial effect against DAD formation. However, it is uncertain whether the reverse or the forward mode inhibition dominates the effect. Data show that, when DADs are the consequence of Na^+^-induced Ca^2+^ overload, it seems feasible that NCX inhibition reduces DAD development, which is presumably the consequence of reverse mode block, and the concomitant decreased Ca^2+^ indirectly reduces the forward mode NCX. This effect resulted in decreased intracellular Ca^2+^ and reduced depolarizing events.

### 4.2. EAD, LQT and TdP

The term EAD indicates a spontaneous depolarizing event that occurs before the completion of repolarization. The classical proposed mechanism for EAD development is the reactivation of the L-type Ca^2+^ current during a long-lasting action potential [51]; however, the roles of the late Na^+^ current [52,53], the NCX [37], CaMKII, and PKA [54] were also suggested as underlying mechanisms. EADs may play an essential role in the arrhythmogenesis by providing arrhythmia triggers that promote re-entry [55]. EADs typically appear when the action potential is excessively long, often due to K^+^ channel inhibition and/or during slow pacing rate. Also, several diseases, such as chronic heart failure or long QT-syndrome, are also reported to be associated with the development of EADs [56]. Torsades de pointes (Tdp) is polymorphic tachycardia which can lead to fibrillation and sudden cardiac death. It is widely accepted that Tdp is initiated by a trigger (EAD or DAD) and maintained by an arrhythmogenic substrate (repolarization dispersion) [57].

In a guinea pig model, LQT3 was pharmacologically mimicked by the enhancement of late Na^+^ current using aconitine, where SEA-0400 was unable to suppress ventricular tachycardia and triggered activity [58]. In another study, SEA-0400 effectively decreased dofetilide-induced TdP arrhythmias in anesthetized dogs in a dose-dependent manner [59]. A lower (0.4 mg/kg) dose of SEA-0400 partially suppressed TdPs, while a higher dose (0.8 mg/kg) completely terminated TdPs; however, the dofetilide-induced QT prolongation remained unchanged [59].

In Langendorff-perfused rabbit hearts with atrioventricular block, NCX inhibition by SEA-0400 did not suppress the incidence of dofetilide-mediated TdP [60].

In contrast, in Langendorff-perfused rabbit hearts, LQT2 and LQT3 were pharmacologically evoked by sotalol or veratridine, respectively [61]. The hearts responded with marked prolongation of monophasic action potential duration, increased dispersion, and, in some cases, TdPs were observed. The administration of SEA-0400 significantly reduced action potential duration as well as the dispersion and TdP incidence. In dog subendocardial tissue, 1 µM SEA-0400 markedly decreased the EAD amplitude [47].

The effect of NCX inhibition on EAD incidence was investigated also in canine right ventricular papillary muscle (Figure 4). EADs were evoked by slow pacing (at 5 s cycle length) and parallel application of 1 µM dofetilide and 100 µM BaCl_2_. It was found that 3 and 10 µM ORM-10103 decreased the amplitude of EADs in a dose-dependent manner, indicating the role of forward mode NCX in the development of EAD. A further (or additional) possibility for the mechanism of action is that ORM increased the intracellular Ca^2+^ level, inducing faster inactivation of Ca^2+^ channels. In this case, if EADs were mediated by Ca^2+^ channels, reopening of the channel performing window Ca^2+^ current—as it was claimed previously in [51]—NCX inhibition might indirectly reduce the amplitude of EADs [37]. In rabbit hearts, it was found that SEA-0400 eliminated the H_2_O_2_-induced EADs [62].

Chang et al. investigated the antiarrhythmic effect of ORM-10103 in normal and failing rabbit hearts [63]. QT-interval prolongation in the ECG curve was stimulated by I_Kr_ inhibition, leading to enhanced EAD development and Tdp inducibility in both control and failing hearts. When ORM-10103 was applied, the number of premature beats was reduced; however, Tdp inducibility remained unchanged. It was found that ORM-10103 increased action potential duration and inhomogeneity in failing hearts. These findings are consistent with previous results by Farkas et al., where SEA-0400 failed to reduce Tdp incidence in rabbit hearts [60]. It is important to note that ORM-10103 is not highly selective: it exerted mild I_Kr_ inhibitory action [37], which could be augmented when repolarization reserve was attenuated, e.g., in the case of heart failure. Therefore, the observed prolongation action potential duration and its increased dispersion is not necessarily related to the NCX blockade.

It seems possible that selective NCX inhibition has a limited effect against EAD development. However, this is probably not surprising since changes in several ion currents, such as I_CaL_, I_NaL_, and I_NCX_ are known to contribute to EAD formation. It is conceivable that the participation of these currents in the EAD formation highly influences the effect of NCX inhibition. Conditions enabling a larger contribution of NCX to EAD development, such as higher cytosolic Ca^2+^ concentration, predict a more effective beneficial influence of selective NCX inhibition on EAD incidence.

### 4.3. Ischemia-Reperfusion

The aerobic metabolism of the heart is seriously compromised during ischemia, which occurs typically after regional coronary artery occlusion. The concomitant decline in intracellular ATP level, intracellular acidosis, deliberation of reactive species, etc., may influence the vast majority of ion channels and transporters [64], leading to marked changes in the intracellular milieu of the cells, action potential configuration, and impulse propagation, ultimately largely increasing the arrhythmia propensity of the heart [65,66,67].

It Is well-known that NCX has a key role in arrhythmogenesis related to ischemia/reperfusion. The rise of the intracellular Na^+^ concentration due to suppression of the Na^+^/K^+^ pump is a consequence of ATP depletion. This may facilitate the reverse mode operation of NCX, leading to net Ca^2+^ gain [68,69,70]. Therefore, it seems feasible that the prognosis of ischemia worsens due to high NCX activity which is supported by murine experiments with heterozygous over-expression of NCX, where increased susceptibility to ischemia/reperfusion injury was observed [71,72].

In vivo studies on ischemia-/reperfusion-related arrhythmias obtained with NCX blockers yielded controversial results, since the majority of experiments failed to show any antiarrhythmic effect of SEA-0400 [73,74,75]. In a one-month myocardial infarction rabbit model, SEA-0400 reduced the number of pacing-induced ventricular premature beats but was also found to be proarrhythmic at the same time due to modifying restitution of action potential duration and enhancing the development of discordant cardiac alternans [76]. Ventricular fibrillation incidence was decreased by NCX inhibition only in the study of Takahashi et al., when studied using a model of coronary occlusion in rats [77].

In Langendorff-perfused rat hearts, SEA-0400 improved the cardiac function and energy metabolism [78] but failed to provide protection against ischemia-/reperfusion-induced ventricular arrhythmias [78,79]. In another study, SEA-0400 was protective against glycolytic inhibition-induced triggered activity in rat hearts; however, this beneficial effect disappeared when the heart rate decreased [80].

The effect of ORM compounds in ischemia/reperfusion injury was investigated on both isolated cells and Langendorff-perfused hearts. Simulated ischemia was induced by a modified external solution containing unusually high K^+^ and lactate concentrations, but no glucose, combined with acidic pH and hypoxia. Intracellular Ca^2+^ level, action potentials, and cell viability were recorded in these experiments.

For single cell investigation, canine left ventricular myocytes were used [81]. Application of the ischemic solution largely increased the diastolic Ca^2+^ level, decreased the amplitude of the Ca^2+^ transient, caused depolarization, shortening, and triangulation of action potentials, and ultimately cell death. An amount of 10 µM ORM-10103 prevented the rise of diastolic Ca^2+^ during both ischaemia and reperfusion (Figure 5), and increased cell survival, while the ischemia-induced changes in action potential morphology were not restored. The marked attenuation of the ischemia-induced rise in diastolic Ca^2+^ suggests that the reverse mode NCX inhibition could be primarily attributable to the observed protective effect. Presumably, the ATP depletion reduced the Na/K ATPase activity and increased intracellular Na^+^ concentration, which drove the reverse NCX activity to overload the cells with Ca^2+^. In line with this, it was also reported that selective NCX inhibition is able to hinder the Na^+^-induced Ca^2+^ gain after strophantin application [34].

Szepesi et al. investigated the possible protective effect of selective NCX inhibition against coronary ligation-induced ischemia in Langendorff-perfused rat hearts, where 10 min of ischemia was followed by 30 min of reperfusion [79]. Extrasystoles, ventricular tachycardia, and ventricular fibrillation were observed during reperfusion. The application of 10 µM ORM-10103 increased the arrhythmia-free period and decreased the incidence of the extrasystoles, while it had no effect on the incidence and duration of ventricular fibrillation.

In contrast, in the case of ischemia-induced by coronary ligation in anaesthetized rats, 1 µM ORM-10962 failed to exert any antiarrhythmic effect [38]. During ischaemia, arrhythmias were not detected, while a large number of life-threatening arrhythmias appeared during reperfusion. In fact, 33% of the animals died by irreversible ventricular fibrillation. Other types of arrhythmias, such as reversible fibrillation or tachycardia, were often observed among survivor animals; however, ORM-10962 failed to cause a change either in the type or the time of onset of the developing arrhythmia.

Similar results were obtained from experiments with zero-flow ischemia in Langendorff-perfused guinea pig hearts. Reperfusion after the no-flow ischemia induced similar ventricular fibrillation in both groups (i.e., control and ORM-10962-treated) without any statistical difference between the results [38].

Although selective NCX inhibition clearly exerts beneficial effects against some pathophysiological components of the ischemia/reperfusion injury, such as the Na^+^-induced Ca^2+^ load, it has a limited protective effect against other symptoms of ischemia/reperfusion injury. It is likely, therefore, that the antiarrhythmic effect observed against the Na^+^-induced Ca^2+^-overload may be attributable mainly to the reverse mode blockade of NCX. However, in a whole heart aspect, several other factors are present that could be unaffected by the NCX inhibitors, such as changes in repolarization, reduction in impulse propagation, depolarization of the resting membrane potential, etc.

### 4.4. Action Potential Duration, Refractory Period, and APD-Dispersion

The equilibrium potential of the NCX predicts that both outward and inward current could be developed during an action potential. However, under normal conditions, the reverse mode is suggested to be restricted in the first section of the action potential [31]; therefore, mainly inward current is expected to be developed. It is predicted that the exchanger activity may lengthen the actions potential duration, thus its inhibition should shorten the action potential and the QT-interval.

Milberg et al. recorded monophasic action potentials from Langendorff-perfused rabbit hearts. When SEA-0400 was applied alone, the drug shortened action potentials within the range of 0.8 and 1.6 µM. Similarly, the QT interval was also shortened in this concentration range [61]. SEA-0400 effectively prevented the action potential lengthening and QT interval prolongation caused by free artemether, an antimalaria agent [82]. In this study, 1 µM SEA-0400 exerted a tendency to shorten action potential in single cells isolated from mice. SEA-0400 also effectively attenuated the repolarization prolongation in a murine model of Chagas disease without shortening it under control conditions [83].

Tadros and Cheung reported no change in the duration of rat action potentials after the down-regulation of NCX [84]. Namekata found unaltered action potential morphology following SEA-0400 treatment [46], similarly to Kormos et al., who also failed to demonstrate any effect of ORM-10103 in canine single cell action potentials [81]. In contrast, Tanaka reported action potential shortening after SEA-0400 in isolated murine myocytes [85]. Nagy et al. also found no change in the action potential configuration, neither in the presence of SEA-0400 nor after ORM-10103 administration. However, it was also reported that SEA-0400 reduced the action potential duration in rabbit isolated cells [62].

In the study where ORM-10962 was used for NCX inhibition, the action potential duration was unaltered after administration of 1 µM of the compound [38]. Slight changes were observed after the manipulation of NCX transport direction: when reverse mode was augmented, action potentials marginally lengthened; when forward mode was facilitated, the plateau potential was slightly depressed.

An important additional effect of selective NCX inhibition is that it increased the effective refractory period [86]. This effect may also contribute to the stabilization of the membrane potential. Action potential dispersion means that differences between the action potentials originated from different regions of the heart. Marked dispersion of action potential duration can be observed between the midmyocardial layer and the endocardial or epicardial myocardium. Similarly, the heterogeneity between the ventricular myocardium and Purkinje fibres is also significant. The excess of dispersion has a crucial role in arrhythmogenesis, since increasing dispersion causes asynchronous refractory periods between different regions, providing a vulnerable window for arrhythmia triggers [87,88]. The effect of NCX-inhibition on dispersion was also tested in some studies.

In the study of Nagy et al., the Purkinje fibre and ventricular muscle action potentials were simultaneously recorded from a Purkinje-ventricular preparation using two microelectrodes, where the two preparations were connected (Figure 6). The application of 10 µM ORM-10103 did not change the dispersion obtained between Purkinje fibre and ventricular action potentials [34].

Milberg et al. demonstrated that the sotalol- or varatridine-induced dispersion between epicardial and endocardial monophasic action potentials was significantly reduced by the application of 1 µM SEA-0400 [89]. This effect was achieved by asymmetrical action potential shortening caused by NCX inhibition.

Human atrial APD_90_ was unchanged when SAR296968, a novel, selective NCX inhibitor compound, was used. Similarly, action potential amplitude and resting potential remained unaltered after selective NCX inhibition (Figure 7) [90].

In summary, the results obtained with NCX inhibition on the action potential duration are controversial. Data obtained from the SEA-0400 experiments showed that it exerted moderate action potential shortening or no effect; however, it cannot be ignored that SEA-0400 also suppressed L-type Ca^2+^ current even at a concentration of 1 µM [36], which could amplify the effect of a potential NCX-inhibition-induced action potential shortening. When using ORM compounds, no effect on action potential duration has been described so far.

### 4.5. Action Potential and Ca^2+^ Transient Alternans

Cardiac alternans refer to a beat-to-beat oscillation of the action potential as well as the intracellular Ca^2+^-transient, appearing typically at high pacing rates. Cardiac alternans is considered a reliable predictor of serious life-threatening arrhythmias such as ventricular fibrillation [91,92]. The underlying mechanisms of alternans are not fully understood. Briefly, there are two leading concepts for the ionic mechanism of alternans: (I.) the voltage-driven theory claims that alternans is expected when the slope of the restitution curve of the action potential duration (i.e., the relation of action potential duration and the preceding diastolic interval) is larger than 1 [93]. (II.) The Ca^2+^-driven theory implies that alternans develops when the beat-to-beat balance of Ca^2+^ handling is compromised [94,95,96]. Since NCX is an important player in intracellular Ca^2+^ handling and may also influence the membrane potential, its inhibition has emerged as a potential strategy to suppress or prevent alternans.

In Langendorff-perfused rat hearts, glycolytic inhibition-related ventricular fibrillation was prevented by 2 µM SEA-0400 in three of six hearts [80]. Simultaneous Ca^2+^ and V_m_ mapping in control rabbits and with one-month myocardial infarct revealed that SEA-0400 suppressed the pacing-induced ventricular premature beats. SEA-0400 steepened the slope of the restitution curve and enhanced spatially discordant alternans. It was concluded that, in one-month infarct hearts, SEA-0400 suppresses pacing-induced ventricular extrasytoles but promotes the development of discordant alternans (Figure 8) [76].

Action potential and Ca^2+^-transients were simultaneously measured in isolated guinea pig ventricular cells. Under control condition positive Ca^2+^-to-V_m_ coupling was found, i.e., large Ca^2+^-transients were associated to long action potentials, indicating the predominance of the inward current generated by NCX, which lengthened the action potentials during Ca^2+^ extrusion. The application of SEA-0400 did not attenuate alternans, but changed the alternans to negative Ca^2+^-to-V_m_ coupling, indicating that L-type Ca^2+^ current became the dominant charge carrier after NCX inhibition. The positive coupling under control conditions suggests the predominance of NCX during action potential alternans [97].

In 2021, the effects of selective NCX inhibition on action potential duration and Ca^2+^-transient alternans were investigated in canine ventricular tissue and isolated myocytes. Both action potential duration and Ca^2+^-transient alternans were evoked by rapid pacing (ranging from 250 to 170 ms cycle length). It was found that 1 µM ORM-10962 significantly decreased the amplitude of both types of alternans, while the slope of the restitution curve remained unaltered. Furthermore, the ORM-10962 increased the refractory period of the action potential. The underlying mechanism of the “anti-alternans” effect was supposed to be a consequence of establishing a new balance in the Ca^2+^ homeostasis when SR Ca^2+^ loading is increased and the SR release refractoriness is reduced. These data also suggest that the main driver of the alternans is the intracellular Ca^2+^ rather than the slope of the restitution curve.

## 5. Inotropic Effects of Novel NCX Inhibitors

NCX has a crucial role in Ca^2+^ extrusion from the cells in each cycle; therefore, its selective inhibition was a promising novel therapeutic tool to modify intracellular Ca^2+^ handling and improve cardiac performance, especially in heart failure. The seminal work of Hobai et al. [98] demonstrated that XIP, a peptide inhibitor of the NCX, was able to restore the reduced Ca^2+^ transient magnitude near to the normal level in canine failing hearts. However, the application of XIP has several limitations. Subsequently, several experimental approaches were performed by using novel inhibitors to address this issue.

Regarding the inotropic action of NCX inhibition, a species-dependent effect seems to be delineated: the results obtained from rat or mice are different from those recorded from larger mammals developing longer, plateau-forming action potentials. In order to better illustrate this discrepancy, the results have to be discussed according to the experimental animal applied.

### 5.1. Selective NCX Inhibition Results in Positive Inotropy in Rat and Mouse and Human Atrium

In 2005, Tanaka et al. investigated the effects of 1 and 10 µM SEA-0400 in rats. An amount of 1 µM SEA-0400 caused about 25% inotropy in tissue preparations, and a similar increase was found in cell shortening and Ca^2+^ transients [85]. Single cell action potential measurements exerted considerable shortening after the application of 1 µM SEA-0400, indicating inward NCX current inhibition; however, this result could be augmented by an additional reduction in L-type Ca^2+^ current.

In 2007, Acsai et al. investigated the effect of selective NCX inhibition in isolated rat myocytes by using 0.3 µM SEA-0400. Ca^2+^ transients and cell shortening were recorded in the absence and presence of SEA-0400 [99]. It was found that both cell shortening (by ~40%) and Ca^2+^ transient amplitude (by ~30%) were increased after selective NCX inhibition. Parallel with increasing the amplitude of the Ca^2+^ transient, the magnitude of L-type Ca^2+^ current decreased. This change could be the consequence of the Ca^2+^ current-inhibitory side effect of SEA-0400; however, additionally, some indirect effect of the increased intracellular Ca^2+^ also could reduce Ca^2+^ current by a Ca^2+^-dependent inactivation process [100].

Similar results were obtained in 2008 by Farkas et al. [101]. In this study, 1 µM SEA-0400 was investigated in Langendorff-perfused rat hearts. Left ventricular performance was tracked by the measurement of developed pressure via non-elastic balloon placed in the ventricle. The application of 1 µM SEA-0400 increased the maximal systolic pressure by 12%. Also, in 2008, Szentandrássy et al. investigated the effect of 0.3 and 1 µM SEA-0400 in Langendorff-perfused rat hearts, recording Ca^2+^ transients [102]. The authors found that both concentrations of SEA-0400 increased the developed pressure and the amplitude of the Ca^2+^ transient. Ozdemir et al. (2008) reported that 1 µM SEA-0400 facilitated cell shortening and decreased the rate of relaxation in murine myocytes [103]. A contradictory result, published in 2015 by Szepesi et al. [79], was obtained from Langendorff-perfused rat hearts, where 1 µM SEA-0400 failed to influence left ventricular systolic pressure.

In 2019, Primessnig et al. investigated the effect of ORM-11035 on heart failure with preserved ejection fraction in rats. It was found that chronic application of the compound markedly attenuated the cardiac remodeling and diastolic dysfunction [104].

In 2022, Hegner et al. used a novel, selective NCX inhibitor SAR296968 on human atrial myocytes [90]. It was found that 3 µM SAR296968 increased the SR Ca^2+^ content and, in 300 nM and in 3 µM, increased the developed tension in human atrial trabeculae.

### 5.2. Selective NCX Inhibition Failed to Exert Positive Inotropy in Guinea Pigs, Rabbits and Dogs

Despite the mostly positive results regarding the inotropic effect of NCX inhibition in rats, studies using guinea pigs, rabbits, or dogs mainly reported that NCX inhibitors failed to influence contractility.

In 2008, Farkas et al. reported no effect of 1 µM SEA-0400 on the maximal systolic pressure of the rabbit heart measured in a Langendorff-perfusion system [101]. This study also reported a positive inotropic effect of SEA-0400 in rat hears; therefore, a direct comparison between the two species was possible, where the experimental conditions were completely identical. Also, in 2008, Birinyi et al. studied the effect of 0.1, 0.3, and 1 µM SEA-0400 on Ca^2+^ transients, cell shortening, and SR Ca^2+^ content in canine myocytes [105] (Figure 9).

In spite of the results of current measurements, indicating that SEA-0400 effectively inhibited the NCX current, no change was found in the transient and cell shortening irrespective of the applied concentration. The results were explained by a Ca^2+^-dependent effect of SEA-0400, suggesting that higher intracellular Ca^2+^ decreases the effect of NCX-inhibition. Therefore, current measurements (where intracellular Ca^2+^ is buffered to a low level) could exert a relatively high (~80%) effect; however, this could be markedly decreased under normal Ca^2+^ handling. Importantly, the relaxation of caffeine-induced Ca^2+^ transient was delayed by 1 µM SEA-0400, suggesting that NCX inhibition could be more pronounced when NCX operates with nearly its full capacity. In line with these results, Szentandrassy et al. (2008) also reported no alteration in the systolic and diastolic pressure and intracellular Ca^2+^ concentration in Langendorff-perfused guinea pig hearts after the application of 0.3 and 1 µM SEA-0400 [102].

A novel NCX inhibitor, ORM-10103, was also tested in canine cardiomyocytes. Under normal conditions, ORM-10103 failed to change the amplitude of the Ca^2+^-transient or the rate of cell shortening. When cells were pre-treated by 2 nM ATX-II (activator of late Na^+^ current), the magnitude of the Ca^2+^-transient was reduced by ORM-10103. Similar results were obtained when the Na/K pump was inhibited by strophantin prior to the application of ORM-10103 [34].

Kohajda et al. studied an improved NCX inhibitor, ORM-10962 (1 µM), in canine ventricular myocytes. In spite of the approximately 80% NCX inhibition in both the forward and reverse modes, ORM-10962 failed to influence the Ca^2+^ transient and cell shortening under normal conditions. In contrast, a positive inotropic effect was observed when intracellular Ca^2+^ was increased by forskolin prior to ORM administration [38]. This result also suggests that, in the case of intact Ca^2+^ handling, the effect of NCX-inhibition may be markedly less than 80%. In order to overcome this, NCX must be considerably burdened by Ca^2+^. In contrast, when reverse NCX was facilitated by a low (70 mM) external NaCl concentration, ORM-10962 reduced the amplitude of Ca^2+^ transients, indicating a reverse NCX-mediated Ca^2+^ loss.

Similar results were obtained in a study by Oravecz et al., where 1 µM ORM-10962 failed to influence the Ca^2+^ transient and cell shortening in canine ventricular myocytes under normal conditions. Augmentation of the reverse mode NCX activity by low external Na^+^ decreased, while facilitation of the forward mode activity increased the effect of NCX inhibition on the amplitude of Ca^2+^ transients [49].

Jin et al. used 3 µM ORM-10103 in porcine atrial myocytes, where 11-deoxycorticosterone acetate was used to induce hypertension. In these cells, it was found that NCX inhibition increased the amplitude of Ca^2+^ transient and also the Ca^2+^ content of the SR in left atrial myocytes but failed to alter the parameters of the Ca^2+^ transient and SR Ca^2+^ content in right atrial cells. This discrepancy could be attributable to the shift of NCX toward reverse mode in the case of right atrial cells. The NCX blockade ratio was found to be approximately 60% in both the left and right atriums [106].

Otsomaa et al. used a brand new ORM compound, ORM-11372, which exerted extremely low EC_50_ values (5 and 6 nM for reverse and forward mode, respectively) for NCX [107]. It was found that ORM-11372 elicited positive inotropy in anaesthetized rabbit left ventricular contractility. Other parameters, such as heart rate or systolic blood pressure, were not influenced.

Pelat et al., in 2021, by using a novel inhibitor, SAR340835, found that selective NCX inhibition improved the systolic function and sympathovagal balance in a dog heart failure model [108].

Taking together inotropic data obtained from SEA-0400, ORM-10103/10962/11372, and SAR296968/340835 experiments, it seems likely that selective NCX inhibition shows a species-dependent effect. In experiments performed in rats (or in human atrial myocytes that exerted very short APD90 (~50 ms)), primarily performed using SEA-0400, mainly positive inotropic effects were found. In contrast, when SEA-0400 or ORM compounds were applied in animal models having long lasting action potentials with a pronounced plateau, the vast majority of the results reported no effect of NCX inhibition.

The most prominent difference between rat and dog/rabbit ventricular cells is probably that the rat action potential is about 3–4 times shorter, lacking the plateau phase. The baseline frequency of the rat heart is also considerably higher than that in dogs or rabbits. The short action potential and the consequent fast decline of membrane potential suggests minimal reverse mode NCX activity during the action potential, since reverse mode exchanger activity is primarily expected at depolarized membrane potential levels. Therefore, it is feasible that, in rats, NCX activity quickly changes to the forward mode when intracellular Ca^2+^ is near to its peak or it is still high; thus, the Ca^2+^ extrusion function is boosted. In this case, NCX inhibition decreases the portion of active exchanger molecules, leading to exhaustion of the Ca^2+^ extrusion capacity of the exchanger that could be manifested in intracellular Ca^2+^ gain. In contrast, in dogs or rabbits, the plateau phase maintains the membrane potential at a depolarized level for a relatively long period of time where the speed of Ca^2+^ extrusion by the NCX is restricted. When the membrane potential drops to more negative values, and forward NCX is boosted, the intracellular Ca^2+^ is in strong decline due to SERCA function. Therefore, the lower intracellular Ca^2+^ may exhaust the NCX capacity less; thus, the NCX-inhibition may be “better tolerated” and unable to shift the Ca^2+^ balance toward higher cytosolic Ca^2+^ levels. However, this assumption could stand for only the compounds that have a limited NCX-inhibitory effect under normal Ca^2+^ cycling (~ 50% [49]). It is feasible that novel compounds having higher potency to inhibit a larger fraction of the NCX could cause positive inotropy in these cases; however, this requires further experiments on large animal models.

## 6. Conclusions

The data suggest that selective NCX inhibition could be a promising novel therapeutic tool against triggered arrhythmias, especially during Na^+^-induced Ca^2+^-overload. In this case, it is feasible that the antiarrhythmic effect is driven by the reverse mode inhibition, mediated by a reduction in cytosolic Ca^2+^ concentration which suppresses triggered potentials induced by the forward mode activity of NCX. The inhibition of NCX could also be promising to attenuate action potential duration and Ca^2+^-transient alternans. However, it is important to note that these ‘beneficial’ effects of NCX inhibition could have important side effects. For instance, reverse mode inhibition could lead to decreased Ca^2+^_I_; however, it may have a detrimental effect to the contraction. Action potential shortening and shifts in the Ca^2+^ handling attenuate alternans; however, they could facilitate the development of atrial fibrillation and increase the propensity of the heart to the development of triggered arrhythmias. These potential unwanted effects could limit the possible future clinical application of NCX inhibitors and require further studies on large animals.

In contrast, selective NCX inhibition seems to be ineffective to induce significant, clinically useful inotropy in ventricular myocardium of larger mammals, including humans. The lack of effect could be based on the peculiar time course of NCX current during the action potential and the relatively low degree of NCX blockade that leaves enough capacity for NCX to maintain normal Ca^2+^ extrusion.

## Figures and Tables

**Figure 1 ijms-23-14651-f001:**
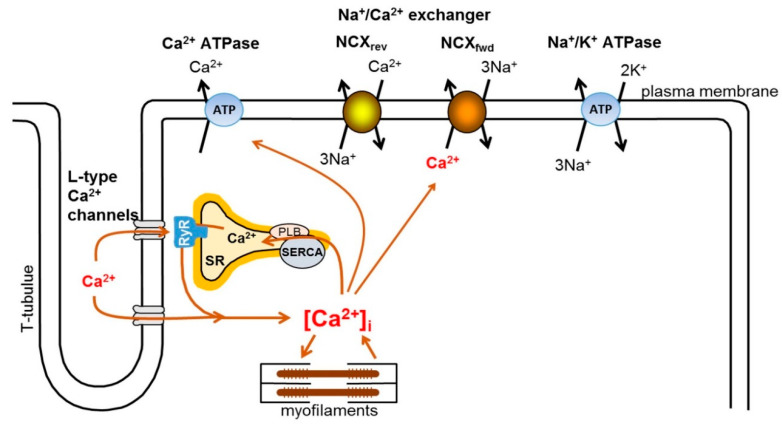
Schematic diagram of intracellular Ca^2+^ handling in ventricular myocytes.

**Figure 2 ijms-23-14651-f002:**
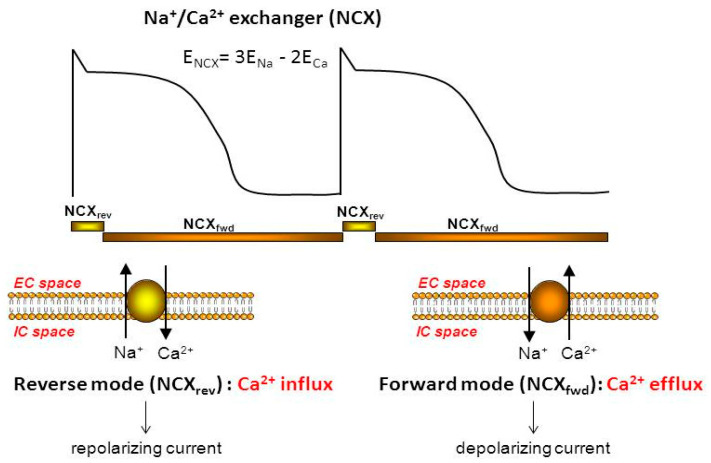
Schematic illustration of the presumed NCX function during ventricular action potential. The reverse mode of the NCX (Ca^2+^ influx/outward current) is restricted to the beginning of the action potential. The forward mode (Ca^2+^ efflux/inward current) is suggested to be active during the latter part of the plateau.

**Figure 3 ijms-23-14651-f003:**
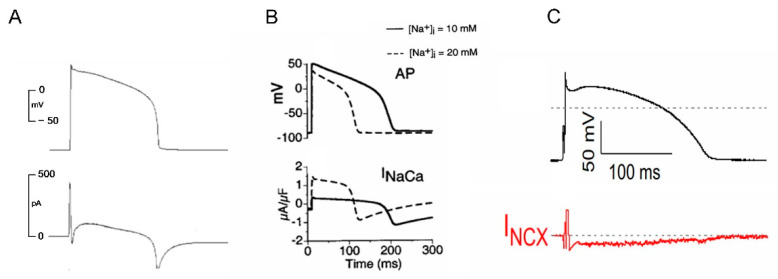
Estimation of NCX current during an action potential. (**A**): Noble et al. (reprinted with permission from Ref. [29]. 2022, Elsevier Science & Technology Journals), the lower panel indicates mainly outward NCX during an action potential. (**B**): Faber and Rudy action potential model demonstrates that NCX current depends on [Na^+^]_i_ (reprinted with permission from Ref. [30]. 2022, Elsevier). (**C**): Horvath et al. defined NCX current during an action potential by using the “onion-peeling” technique in dog ventricular myocytes. (reprinted with permission from Ref. [35]. 2022, Elsevier).

**Figure 4 ijms-23-14651-f004:**
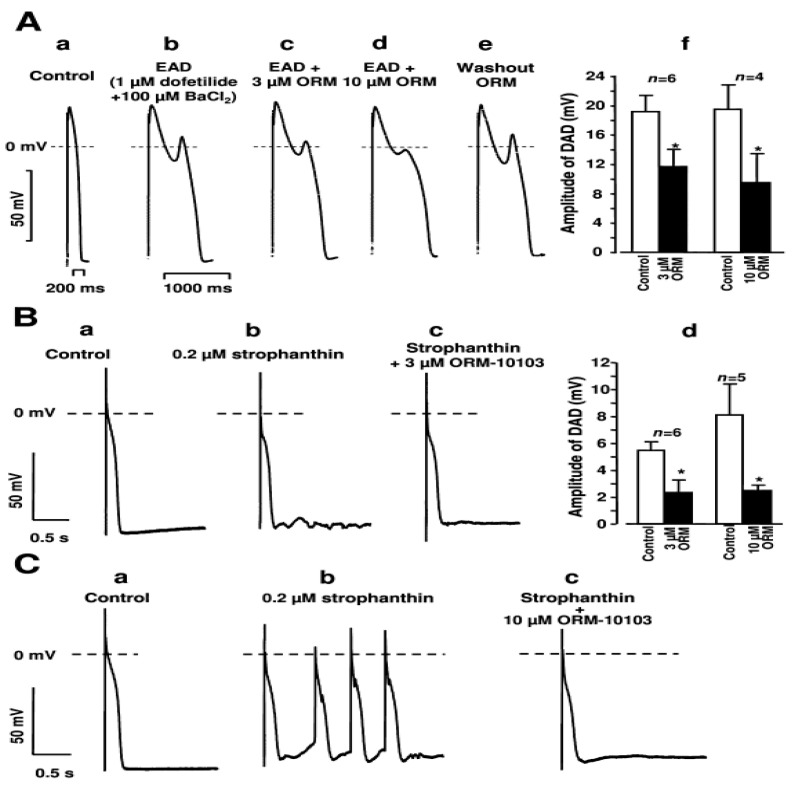
Effect of ORM-10103 against triggered activity. (**A**) ORM-10103 reduces the amplitude of EAD. (**a**–**e**) indicates the subsequent sections of the experiment. Bar graphs show the effect of ORM on EAD amplitude in 3 and 10 µM (**f**). (**B**) ORM-10103 decreases the strophantin-induced membrane potential oscillations. (**a**–**c**) indicates the subsequent sections of the experiment, and bar graphs show the numerical data of the result (**d**). (**C**) ORM-10103 reduces the incidence of DADs. (**a**–**c**) indicates the subsequent sections of the experiment. * denotes *p* < 0.05. Reprinted with permission from Ref. [37]. 2022, Wiley.

**Figure 5 ijms-23-14651-f005:**
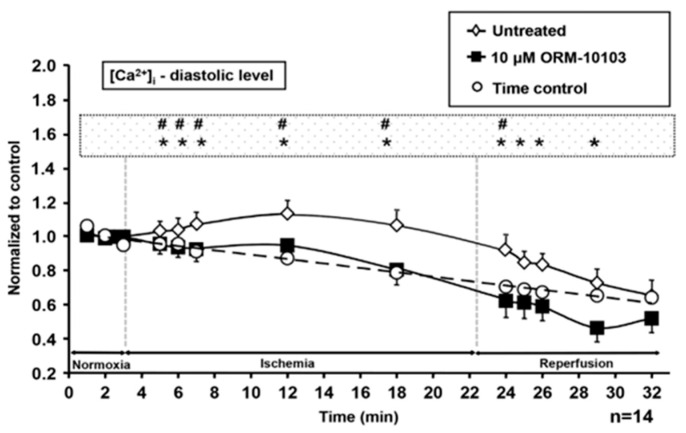
Selective NCX inhibition by 10 µM ORM-10103 avoids diastolic Ca^2+^ increase during simulated cellular ischemia in canine isolated ventricular myocytes. # and * indicate significant (*p* < 0.05) difference between time control vs. untreated and untreated vs. ORM-treated values, respectively. Reprinted with permission from Ref. [81]. 2022, Elsevier.

**Figure 6 ijms-23-14651-f006:**
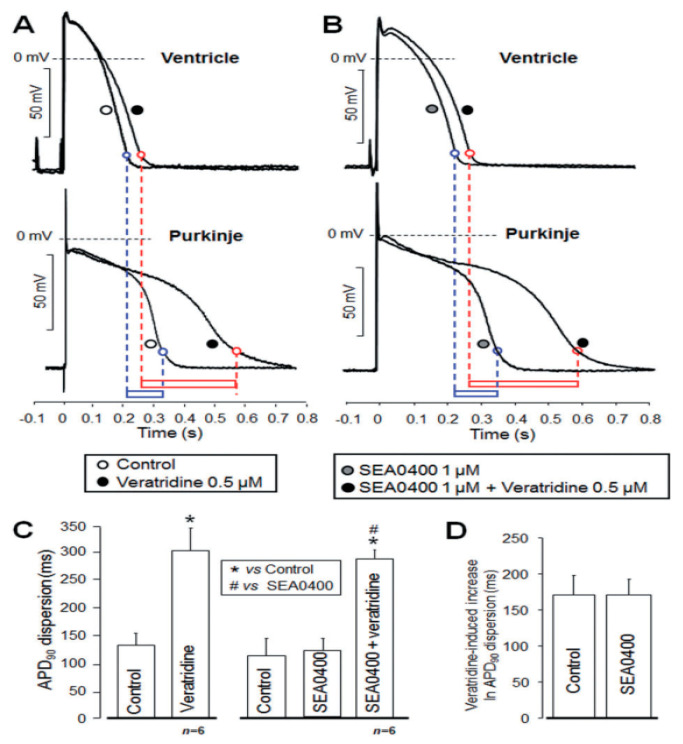
Selective NCX inhibition does not influence the ventricle–Purkinje action potential dispersion. (**A**): after control recordings (open symbol) 0.5 µM veratridine (filled symbol) was applied. (**B**): Prior to verartidine application 1 µM SEA-0400 was employed. Bar graphs illustrate the APD dispersion (**C**) and the veratridine induced increase of APD dispersion under control condition and in the presence of SEA-0400 (**D**). Reprinted with permission from Ref. [34]. 2022, Wiley.

**Figure 7 ijms-23-14651-f007:**
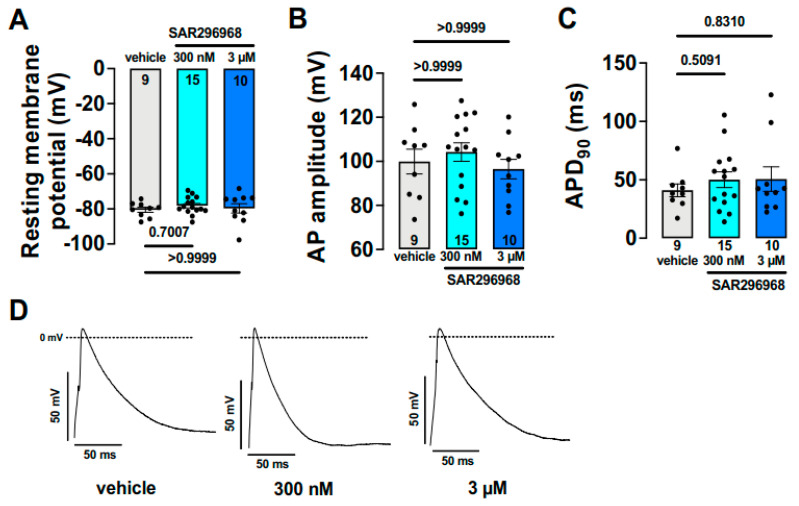
Effect of SAR296968 on human atrial action potentials. Bar graphs (**A**–**C**) and original recordings (**D**) indicate no effect of NCX inhibition on action potential duration.

**Figure 8 ijms-23-14651-f008:**
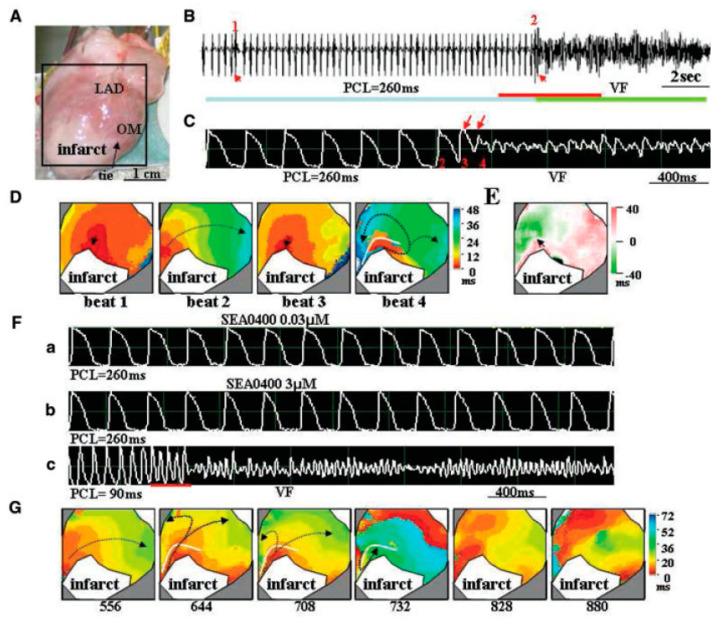
Effect of SEA-0400 on ventricular arrhythmias measured by optical mapping. (**A**): Mapped region; (**B**): ECG measurement. Red arrows indicate ventricular premature beats; Numbers indicate the number of beats (**C**): action potentials corresponding from the red bar in panel. Red arrows indicate ventricular premature beats; (**D**): isochronal map showing beat marked in (**B**) (beat 1) and in panel (**C**) (beat 2–4); (**E**): APD_80_ map with nodal line; (**F**): action potential tracing: a and b—no arrhythmia was observed in the presence of 0.03 and 3 µM SEA-0400 at 260 ms pacing cycle length c—at 90 ms pacing cycle length fibrillation was observed in the presence of 3 µM SEA-0400; (**G**): isochronal map corresponding to (**F**c). Numbers indicate the time of data acquisition. Frames 708 and 732 illustrate formation of re-entrant wavefront, 828 and 880 depict multiple wavefronts (Reprinted with permission from Ref. [76]. 2022, Wiley).

**Figure 9 ijms-23-14651-f009:**
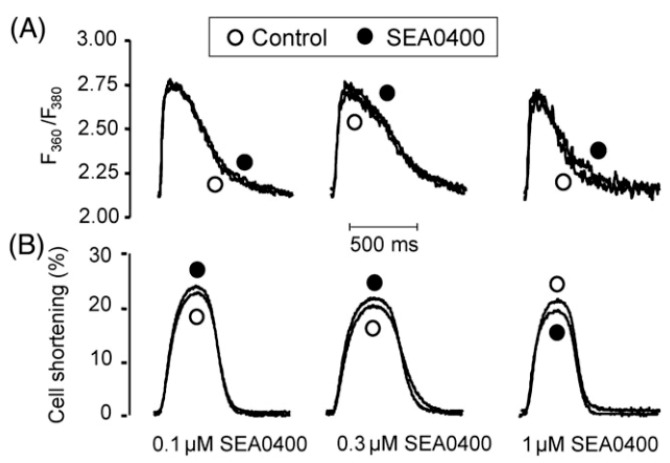
SEA-0400 failed to influence the Ca^2+^ transient (**A**) and cell shortening (**B**) amplitudes in canine isolated myocytes. Reprinted with permission from Ref. [105]. 2022, Oxford University Press).

## Data Availability

Not applicable.

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
