# Peer review of "Antiarrhythmic and Inotropic Effects of Selective Na+/Ca2+ Exchanger Inhibition: What Can We Learn from the Pharmacological Studies?"

_ijms, 2022, doi:10.3390/ijms232314651_

Round 1

Reviewer 1 Report

Please read the reviewer's comments in the attached file (highlighted lines contain corresponding comments/questions) and modify the manuscript according to the comments.

Authors should carefully check the entire text for proper and consistent use of abbreviations and names, grammar (in a few cases) and punctuation.

Figures were provided with low quality and should be used with high pixel resolution in a final publication. In some cases the original images seem to be disproportionally scaled and therefore have improper letter scales; this also is highly recommended for improvement.

Author Response

The authors would like to thank to the Reviewer for the thorough and helpful revision. Please find the answers of the authors in the attached pdf document surrounded with diamond operators below your comments.

Other changes: The Figure 3, 7 and 8 were changed because of the discrepancy between the WoltersKluwer (Circulation Research  and Heart Rhythm journals) and MDPI publication policy.

Reviewer 2 Report

interesting topic, not only for scientific reasons, but also, I hope, for practical reasons soon. Personally, I like the work very much, but I have a few comments, the introduction of which will increase its quality:
 The introduction is short, the reader quickly finds the main topic of the work, and it is not localized in the current therapeutic assumptions.
For this reason, I propose to add a section on inotropic treatment to the introduction. Please pay attention to the classic digoxin - with regard to the drug at the last ESC congress, there were a lot of reports higher than previously positioning the drug. There are also clinical and observational studies. The latter are interesting because they will allow for the evaluation of the concentration optimization in the therapy monitored with digogxin. up to 20 years of observations are available.
In the introduction, it would also be advisable to add a section on the use of inotropically positive drugs with an effect not related to calcium concentration, e.g. levosimendan (it is also worth paying attention to repetitive use).

Author Response

The authors would like to thank to the Reviewer for the thorough and helpful revision.

interesting topic, not only for scientific reasons, but also, I hope, for practical reasons soon. Personally, I like the work very much, but I have a few comments, the introduction of which will increase its quality:
 The introduction is short, the reader quickly finds the main topic of the work, and it is not localized in the current therapeutic assumptions.
For this reason, I propose to add a section on inotropic treatment to the introduction. Please pay attention to the classic digoxin - with regard to the drug at the last ESC congress, there were a lot of reports higher than previously positioning the drug. There are also clinical and observational studies. The latter are interesting because they will allow for the evaluation of the concentration optimization in the therapy monitored with digogxin. up to 20 years of observations are available.
In the introduction, it would also be advisable to add a section on the use of inotropically positive drugs with an effect not related to calcium concentration, e.g. levosimendan (it is also worth paying attention to repetitive use).

Thank you for this comment. The Introduction section was supplemented accordingly.

Other changes: The Figure 3, 7 and 8 were changed because of the discrepancy between the WoltersKluwer (Circulation Research  and Heart Rhythm journals) and MDPI publication policy.

Reviewer 3 Report

In this manuscript Nagy et al write about the antiarrhythmic and inotropic effects of selective Na+/Ca2+ exchanger inhibition. The topic goes deep within the subfield of pharmacology of cellular electrophysiology in the heart. In such a specific topic, I find it important to relate the issues to a larger frame, and to discuss the matters also on a larger frame. For example:

Line 213. “NCX inhibition is able to set a new Ca2+ balance where NCX activity is reduced and intracellular Ca2+ is elevated, leading to consequent increase in contractility.”

Elevation of intracellular calcium could also, among other things, then lead to decreased/prolonged relaxation (diastolic heart failure), and in the long term the development of an anatomical substrate for arrhythmia via mitochondrial calcium overload > apoptosis > fibrosis.

This really is a “chicken and egg” issue. The increase and decrease of NCX activity can both be listed to have negative or positive effects on arrhythmia, inotropy, fibrosis, contraction, relaxation etc. It seems that you can’t have it all, achieving a beneficial effect somewhere will likely cause detrimental effects elsewhere.

Electrophysiology is so complex and dependent of numerous variables and conditions, that standardizing as many of those is crucial. What becomes clear is the importance of studying these questions with the right models (large animals, humans), having the right anatomical substrate (large heart with similar to human electrophysiology). This complexity is well demonstrated by reference 88 on line 619.

As the conclusion states, “Inhibition of NCX could be also promising to attenuate action potential duration and Ca2+-transient alternans.” This is another example demonstrating the complexity of the issues and the conclusions drawn. On the other hand, decreasing APD alternans would need shortening of APD, which would likely lead to other problems, eg. increased incidence of atrial fibrillation, and attenuating calcium alternans might increase triggered arrhythmias via shortened refractoriness (increased SERCA and/or RyR2 function). With these matters there is always the other side of the coin. Often improving systolic and/or diastolic function seems to increase arrhythmia probability and vice versa.

Please discuss all these issues from the viewpoint of a larger frame, bringing forth the complexity of the topic and the need for experimental studies on large mammals and the for longer periods of time.

On line 62, I wouldn’t state that “ALL Ca2+ released from the intracellular store is the consequence of the trigger Ca2+ entering mainly via L-type Ca2+ channels” Think of RyR2 channels in CPVT.

Please check lines 68-9: “stored Ca2+ is released from discrete units from the sarcoplasmic reticulum which are called Ca2+ sparks.” Calcium sparks are the calcium release events, not the discrete SR units.

Line 105: “ion” should be “in”.

Line 239: “SEA-0400 was reported to effectively suppress arrhythmogenic Ca2+ releases and afterdepolarizations”. Was it really found to suppress the calcium releases themselves? Or just the afterdepolarizations (due to the calcium releases)?

I find the conclusion right, to focus more on inhibiting afterdepolarizations as opposed to improving contractile function.

Author Response

The authors would like to thank to the Reviewer for the thorough and helpful revision.

In this manuscript Nagy et al write about the antiarrhythmic and inotropic effects of selective Na+/Ca2+ exchanger inhibition. The topic goes deep within the subfield of pharmacology of cellular electrophysiology in the heart. In such a specific topic, I find it important to relate the issues to a larger frame, and to discuss the matters also on a larger frame. For example:

Line 213. “NCX inhibition is able to set a new Ca2+ balance where NCX activity is reduced and intracellular Ca2+ is elevated, leading to consequent increase in contractility.”

Elevation of intracellular calcium could also, among other things, then lead to decreased/prolonged relaxation (diastolic heart failure), and in the long term the development of an anatomical substrate for arrhythmia via mitochondrial calcium overload > apoptosis > fibrosis.

This really is a “chicken and egg” issue. The increase and decrease of NCX activity can both be listed to have negative or positive effects on arrhythmia, inotropy, fibrosis, contraction, relaxation etc. It seems that you can’t have it all, achieving a beneficial effect somewhere will likely cause detrimental effects elsewhere.

Electrophysiology is so complex and dependent of numerous variables and conditions, that standardizing as many of those is crucial. What becomes clear is the importance of studying these questions with the right models (large animals, humans), having the right anatomical substrate (large heart with similar to human electrophysiology). This complexity is well demonstrated by reference 88 on line 619.

As the conclusion states, “Inhibition of NCX could be also promising to attenuate action potential duration and Ca2+-transient alternans.” This is another example demonstrating the complexity of the issues and the conclusions drawn. On the other hand, decreasing APD alternans would need shortening of APD, which would likely lead to other problems, eg. increased incidence of atrial fibrillation, and attenuating calcium alternans might increase triggered arrhythmias via shortened refractoriness (increased SERCA and/or RyR2 function). With these matters there is always the other side of the coin. Often improving systolic and/or diastolic function seems to increase arrhythmia probability and vice versa.

Please discuss all these issues from the viewpoint of a larger frame, bringing forth the complexity of the topic and the need for experimental studies on large mammals and the for longer periods of time.

We agree with the Reviewer. The Conclusion chapter was supplemented accordingly. 

On line 62, I wouldn’t state that “ALL Ca2+ released from the intracellular store is the consequence of the trigger Ca2+ entering mainly via L-type Ca2+ channels” Think of RyR2 channels in CPVT.

We agree with the Reviewer. The text was modified.

Please check lines 68-9: “stored Ca2+ is released from discrete units from the sarcoplasmic reticulum which are called Ca2+ sparks.” Calcium sparks are the calcium release events, not the discrete SR units.

We agree, and this section was completely reformulated.

Line 105: “ion” should be “in”.

Corrected

Line 239: “SEA-0400 was reported to effectively suppress arrhythmogenic Ca2+ releases and afterdepolarizations”. Was it really found to suppress the calcium releases themselves? Or just the afterdepolarizations (due to the calcium releases)?

Thank you for this point. Indeed, the text is misleading since these studies did not measure Ca-signals. The text was modified accordingly. However, the NCX-block indeed able to suppress spontaneous Ca-release events (initiated by strophantine). This result is obtained from ref.No.#34. It is discussed in 4.1 subchapter in line 347-353.

I find the conclusion right, to focus more on inhibiting afterdepolarizations as opposed to improving contractile function.

The authors thank again for the work of the Reviewer.

Other changes: The Figure 3, 7 and 8 were changed because of the discrepancy between the WoltersKluwer (Circulation Research  and Heart Rhythm journals) and MDPI publication policy

Reviewer 4 Report

I have reviewed the review manuscript by Nagy Nobert et al. entitled "Antiarrhythmic and inotropic effects of selective Na+/Ca2+ exchanger inhibition: what can we learn from the pharmacological studies?”

In this review, the authors mentioned the physiological importance of Na+/Ca2+ exchanger (NCX) based on its function of cardiac calcium handling. And then they summarized the experimental results of the NCX inhibition focusing on their data obtained by novel highly selective inhibitors. They concluded that the selective inhibition of NCX may be a promising therapeutic tool against triggered arrhythmias especially during Na+-induced Ca2+-overload but does not appear to induce clinically useful inotropic effects in the ventricles of larger mammals, including humans.

In general, the antiarrhythmic and inotropic effects of selective NCX inhibition are very important issues in the therapeutic strategy for heart failure.

To further improve this review, I suggest the following.

Line 80, NCX extrudes 3 Na+ for 1 Ca2+ is it correct?

I think it’s important to mention and discuss the therapeutic potency of NCX inhibitors, citing the following recent important papers.

Biomedicines 2022 Aug; 10(8): 1932.doi10.3390/biomedicines10081932

SAR296968, a Novel Selective Na+/Ca2+Exchanger Inhibitor, Improves Ca2+Handling and Contractile Function in Human Atrial Cardiomyocytes

J Pharmacol Exp Ther2021 377(2):293-304. doi: 10.1124/jpet.120.000238.

SAR340835, a Novel Selective Na+/Ca2+Exchanger Inhibitor, Improves Cardiac Function and Restores Sympathovagal Balance in Heart Failure

Author Response

The authors would like to thank to the Reviewer for the thorough and helpful revision.

I have reviewed the review manuscript by Nagy Nobert et al. entitled "Antiarrhythmic and inotropic effects of selective Na+/Ca2+ exchanger inhibition: what can we learn from the pharmacological studies?”

In this review, the authors mentioned the physiological importance of Na+/Ca2+ exchanger (NCX) based on its function of cardiac calcium handling. And then they summarized the experimental results of the NCX inhibition focusing on their data obtained by novel highly selective inhibitors. They concluded that the selective inhibition of NCX may be a promising therapeutic tool against triggered arrhythmias especially during Na+-induced Ca2+-overload but does not appear to induce clinically useful inotropic effects in the ventricles of larger mammals, including humans.

In general, the antiarrhythmic and inotropic effects of selective NCX inhibition are very important issues in the therapeutic strategy for heart failure.

To further improve this review, I suggest the following.

Line 80, NCX extrudes 3 Na+ for 1 Ca2+ is it correct?

However, there are some debate regarding the exact stoichiometry of the NCX, the widely accepted ratio is 3 Na and 1 Ca. Please see this review: https://pubmed.ncbi.nlm.nih.gov/11805843/

I think it’s important to mention and discuss the therapeutic potency of NCX inhibitors, citing the following recent important papers.

Biomedicines 2022 Aug; 10(8): 1932.doi10.3390/biomedicines10081932

SAR296968, a Novel Selective Na+/Ca2+Exchanger Inhibitor, Improves Ca2+Handling and Contractile Function in Human Atrial Cardiomyocytes

J Pharmacol Exp Ther2021 377(2):293-304. doi: 10.1124/jpet.120.000238.

SAR340835, a Novel Selective Na+/Ca2+Exchanger Inhibitor, Improves Cardiac Function and Restores Sympathovagal Balance in Heart Failure

Thank you for these papers. The results of the mentioned studies were incorporated into the manuscript.

Other changes: The Figure 3, 7 and 8 were changed because of the discrepancy between the WoltersKluwer (Circulation Research  and Heart Rhythm journals) and MDPI publication policy.

Round 2

Reviewer 2 Report

after the implementation of the corrections, the manuscript seems to be ready for publication